# Juvenile Hormone Involved in the Defensive Behaviors of Soldiers in Termite *Reticulitermes aculabialis*

**DOI:** 10.3390/insects15020130

**Published:** 2024-02-14

**Authors:** Yiying Li, Letong Yin, Ruiyao Guo, Yunliang Du, Bo Wang, Long Liu, Zhenya Li, Wei Liu, Guozhi Zhang, Shiheng An, Xinming Yin, Lijuan Su

**Affiliations:** 1College of Life Science, Henan Agricultural University, Zhengzhou 450002, China; ryujje@163.com (Y.L.); 18618107171@163.com (L.Y.); guoruiyao999@163.com (R.G.); yunliangdu@163.com (Y.D.); 13253392364@163.com (B.W.); liuv@henau.edu.cn (W.L.); guozhi_zhang@126.com (G.Z.); 2College of Plant Protection, Henan Agricultural University, Zhengzhou 450002, China; lliu1988@henau.edu.cn (L.L.); zhenya0371@163.com (Z.L.); anshiheng@aliyun.com (S.A.)

**Keywords:** termite, juvenile hormone, JHA-feeding experiment, defensive behaviors, soldier caste-specific protein 1, social behaviors modulation

## Abstract

**Simple Summary:**

Soldiers and workers collectively perform defensive behaviors in termite colonies. To determine the molecular mechanism driving the defensive behaviors of termite soldiers, we tested the effect of the juvenile hormone analogue pyriproxyfen (JHA) and the soldier caste-specific protein 1 (RaSsp1) on the defensive behaviors of *Reticulitermes aculabialis* soldiers when encountering an ant invasion. The results showed that JHA-feeding may affect defensive behaviors (i.e., bites and head-banging) of soldiers via downregulation of *RaSsp1* expression. Feeding JHA may also affect the content of alarm pheromones in soldiers, including limonene, which also influenced the defensive behavior of soldiers. These results suggest that JH and RaSsp1 may play important roles in modulating social defense in termite colonies.

**Abstract:**

Eusocial insects have evolved specific defensive strategies to protect their colonies. In termite colonies, soldiers perform a colony-level defense by displaying mechanical biting, head-banging and mandible opening–closing behaviors. However, few studies have been reported on the factors modulating defensive behaviors in termites. Owing to JH (juvenile hormone) being involved in soldier differentiation, JH was speculated to affect defensive behaviors in termite soldiers. To determine the effect of JH on the defensive behaviors of termite soldiers, we performed a JHA-feeding and *RaSsp1*-silencing experiment and then tested the changes in defense-related behaviors, alarm pheromones and key JH signaling genes. The observed result was that after feeding workers with JHA, soldiers displayed the following: (1) decreased biting events and increased head-banging events; (2) a reduced expression of *RaSsp1* and increased expression of *Met* (methoprene-tolerant, the nuclear receptor of JH) and *Kr-h1* (the JH-inducible transcription factor Krüppel homolog 1); and (3) a decreased concentration of alarm pheromones, including α-pinene, β-pinene and limonene (+, −). Further study showed that soldiers silenced for *RaSsp1* also exhibited (1) decreased biting events and increased head-banging events and (2) increased expression of *Met* and *Kr-h1*. In addition, soldiers stimulated by the alarm pheromone limonene displayed an increase in the frequency of mandible opening–closing and biting behavior. All of these results show that JHA influenced the defensive behaviors of termite soldiers, possibly via downregulating RaSsp1 expression, up-regulating Met and Kr-h1 and stimulating the secretion of alarm pheromones, suggesting that the JH pathway plays important roles in modulating social behaviors in termite colonies.

## 1. Introduction

To protect colonies from omnipresent threats such as predators, termites have evolved diverse defensive strategies, including mechanical and chemical defenses. The mechanical defense includes locomotion, mandibular threats, head-banging, tremulation, touching and grooming [1,2,3]. Due to their specialized mandibles, termite soldiers are more focused on defensive functions than workers in the colony [4]. The specialized mandibles of soldiers serve as effective weapons for defending against enemies. With their high-speed burst attack, they can bite and even kill predators such as ants, effectively defending the colony [5,6]. In nature, when encountering a predator, soldiers exhibit defensive behaviors more frequently than workers do, despite both termite soldiers and workers attacking the ants. Soldiers directly bite off the ant’s antennae, legs or abdominal segment and rarely exhibit an escape response. In contrast, workers exhibit an escape response and are far away from the predators, occasionally attacking the predators [7,8]. In lab colonies, soldiers exhibit a patrolling state, walking around the outside of a Petri dish in the arena accompanied by other defensive behaviors such as head-banging, longitudinal oscillatory movement, antennation, and mandible opening–closing. The first three behaviors were considered to transmit alarm messages to other termites [9], such as head-banging which results in vibrations in the substratum, triggering escape among workers and thereby acting as an alarm communication signal that spreads throughout the whole group via positive feedback [10], and mandible opening–closing for releasing defensive pheromones from the head of the soldier [11]. These diversely defensive strategies result from a plethora of evolutionary changes to increase the collective survivorship of the individuals participating in communal life [3,11]. The reports on the molecular mechanisms driving defensive behavior in termite colonies are growing in number, but most of the studies focus on the worker and few studies are reported on the soldier [12,13].

In contrast to the mechanical defense, the chemical defense involves in the release of a volatile substance—an alarm pheromone from the soldiers’ frontal gland. These alarm pheromones warn conspecifics of danger and provoke strong dose- and context-specific responses, resulting in subsequent adaptive modifications [14]. These modifications include a general increase in the activity level, changes in locomotive, retreat, fight, and the recruitment of other termites to the site of the disturbance in their behavior [3]. Among some species soldiers (*Nasutitermes*, *Veloeitermes velox* and *Reticulitermes aculabialis*), the frontal secretion contains a wide range of compounds including various terpenoids (mono-, sesqui-, and diterpenes), with α-pinene and limonene causing alarm propagation [14,15,16].

As one of the most phenotypically plastic insects, the soldier caste fate in termites could be changed during postembryonic development. That is, a constant high juvenile hormone (JH), or the juvenile hormone analogue pyriproxyfen (JHA), titer triggers soldier differentiation from workers through two molting periods via a pre-soldier stage (that is, worker–pre-soldier–soldier) [4,17]. During this differentiation process, the allometry of soldier’s head occurs, especially regarding the exaggeratedly enlarged mandibles, which are special weapons of soldiers’ defense [18,19,20,21,22]. Moreover, the soldier differentiation could be affected by several up- and downstream genes in the JH signaling pathway, for example, *Met* (Methoprene-tolerant, the nuclear receptor of JH) and *Kr-h1* (the JH-inducible transcription factor *Krüppel homolog* 1) [21]. The effect of JH on defensive behaviors has been reported in several insect species, in which high JH titers cause increased male aggression in lobster cockroaches, *Nauphoeta cinerea* [23], and in the paper wasp *Polistes gallicus*, JH has been found to significantly influence the likelihood of female dominance in the treated group [24]. Among honey bee colonies, JH is one of the factors that affect honeybee aggression [25]. These studies only focused on the phenotype and molecular mechanism of JH regulating the caste differentiation of soldiers, so JH was speculated to affect the defensive behaviors in termite soldiers by regulating the positive expression of Met and Kr-h1 and the release of a volatile substance—an alarm pheromone. It is interesting and worthwhile to clarify that JH further regulates the related behavioral phenotypes and molecular mechanisms of soldiers after caste differentiation, for this hypothesis has been confirmed in eusocial Hymenoptera [26].

Previous studies have shown that RaSsp1, a soldier caste-specific protein in *Reticulitermes aculabialis*, is homologous with the JH-binding protein and is extremely highly expressed in the soldier caste [17]. Silencing *RaSsp1* leads to a shortened head capsule, reduced mandible size, delayed molting time, and decreased molting rate at the beginning of worker or pre-soldier gut-purging [17]. Based on the above research findings, we propose the following hypothesis. JH, which affects the caste differentiation of soldiers, may also influence their attack behavior. RaSsp1 exhibits homology with the JH-binding protein and is potentially influenced by JH, subsequently affecting the defensive behavior of soldiers. This influence is analogous to the role of JH in its biosynthetic pathway.

However, the effect of *RaSsp1* on the defensive behavior of soldiers has not been reported. Here, we tested the effect of JH on the defensive behavior of *R. aculabialis* soldiers via JHA-feeding or the injection of *RaSsp1* dsRNA and then determined the expression changes in the key genes (*Met, Kr-h1* and *RsSsp1*) in JH signal transduction. These results may allow us to understand that JH acts on individual responsiveness to threats and may at least partially explain the ecological success of these eusocial insects.

## 2. Materials and Methods

### 2.1. Insect Collection and JH Treatment

Termites *R. aculabialis* were collected from Huangbai Mountain (31°57′ N, 115°21′ E), Shangcheng County, Henan province, China. Ants *Tetramorium caespitum* were collected from a tree garden (34°67′ N, 113°57′ E) in Zhengzhou city, Henan province, China. The termites include five colonies from different locations. The colonies were kept in a constantly dark room at a temperature of approximately 25 °C and humidity at 70%. Each colony was placed independently. All termites lived in the wood they originally lived in in the wild and could freely move around in their original nests, with sufficient wood and water sources.

To analysis the effect of JH on the soldier’s defensive behaviors, there was minor treatment changes in JHA (Pyriproxyfen, Sigma, Steinheim, Germany) fed to the termites. According to the previous reports, the purity of pyriproxyfen is 99% [17]. The induction rate of 120 μg pyriproxyfen-treated soldiers is like that of 200 ug, but the mortality rate is quite different [27]. To understand the effects of the two treatments on the behavior of soldiers, we designed the following experiment. The details are as follows: after termites were starved for 24 h, 120 or 200 μg JHA (600 μL acetone using a solvent) was uniformly sprayed on the filter paper to feed termites, and the acetone as the control. According to the ratio of worker to soldier at 3:1 when termites go out to forage, termites were cultured in a Petri dish with JHA filter paper at 25 °C and 70% humidity in the dark. The diameter of the filter paper was 7 cm, which is the same as the bottom of the Petri dish, allowing the soldiers to ingest pyriproxyfen both through trophallaxis and by contact with treated surfaces. The use of pyriproxyfen was uniformly handled at the beginning of the experiment, with samples taken over 5 days and the results recorded over time. Because the effect of JH on gene expression returns to near control levels by day 4 [27], and considering possible delays in protein expression and behavioral responses, the experiment was delayed by 1 day for a total of 5 days. Therefore, 270 soldiers were collected in each day from the 1st day to the 5th day for the following experiment, including three groups: 60 for the analysis of defensive behaviors of termite soldiers to an ant, 90 for the gene expression of JH signal key genes and 120 for the analysis of pheromone content in the head of soldiers. Within 5 days after the start of the experiment, every day at 3 pm, freshly treated soldiers were collected for testing.

### 2.2. The Effects of Defensive Behavior on the Enemy Ants

In order to measure the defensive behavior of termites under natural conditions, we adjusted a detecting device according to Ishikawa and Miura (2012) [28]. The details are as follows: a plastic Petri dish with a diameter of 30 mm was used as an artificial nest (Figure 1A), which used one soldier and three workers for more than 20 min for termites acclimating to the artificial nest. The entrance of the device was temporarily sealed with plastic wrap to prevent the activity of termites outside the device. One ant’s (*T. caespitum*) abdomen was glued to the tip of the toothpick and placed in the entrance, so that it could move its head and thorax freely. A digital video camera SONY HDR-CX450 (Suoguang Electronics Co., Ltd., Shanghai, China) was placed above the device to record defensive behaviors, including bites, head-banging, regular opening–closing of the mandibles and antennation [28]. The number of defensive behaviors in the treatment groups (fed with JHA) and the control group (fed with acetone) was counted within 5 min. Furthermore, 20 soldiers were used as biological replicates, and each replicate included three treatments (120, 200 μg JHA and control) and 5 time points. The 20 soldiers were obtained from 5 colonies, with 4 soldiers taken from each colony.

### 2.3. Quantitative Reverse Transcription PCR

To analyze the effect of JH on the JH signal key gene in soldiers, Quantitative Reverse Transcription PCR was performed using the sample of the soldiers’ head on the 1st to 4th day after feeding JHA. Met, Kr-h1 and RaSsp1 were selected as JH key genes. The detail of qRT-PCR was the same as in a previous report [17]. RNA was extracted from 5 heads of soldiers with Trizol RNA extraction reagent (Thermo, Carlsbad, CA, USA), and then cDNA was obtained using the reverse transcription kit HiScript III RT SuperMix for qRT-PCR (Vazyme Bio, Nanjing, China). qRT-PCR was performed using an Applied Biosystems 7500 Fast Real-Time PCR system (ABI, Foster City, CA, USA). Ribosomal protein gene L13a (RPL13a) and elongation factor 1α (EF1-α) mRNA levels were used as references. The specific primers were as follows, *RaSsp1*: F 5′-ACTGTGCTTGGCGCTGTC-3′, R 5′-CTGGGATGTGGTATTGCTT T-3′; *Met*: F 5′-GCCCTCATCATCCGCCTT, R 5′-CTTGCCATCACGAGAACG-3′; *Kr-h1*: F 5′-AGCAGCCCAGATTTACCT, R 5′-GTCTTCGCCCTCCTTTCC-3′; *RPL13a*: F 5′-TCTGTGGAGGACGGTTAG, R 5′-ACTTTCTGCCTGGTTTCA-3′; and *EF1-alfa*: F 5′-CCCTTCGTCTTCCTCTTC, R 5′-CTCCAGCGACATAACCAG-3′. Each experiment was performed with three biological replicates, and three qRT-PCR analyses were performed using the same cDNA sample of each time point.

### 2.4. RNAi Experiment

*RaSsp1* RNAi was used to analyze the effect of *RaSsp1* on the defensive behavior. Firstly, ds RNA of *RaSsp1* was synthesized via the same method used in a previous report [17]. The partial cDNA sequences of *RaSsp1* genes were amplified using gene-specific primers (*RaSsp1*: F 5′-AGTGCTCGATCCGCTGTA-3′; R 5′-GCGCCAGTTCTCGTTTAT-3′; *Egfp*: F 5′-ATGGTGAGCAAGGGCGAGG-3′, R 5′-TTACTTGTACAGCTCGTCCAT G-3′). The amplified product was subcloned into the pMD18-T Vector and were transferred into DH5α competent cells (Sangon Biotech, Shanghai, China). The bacteria were cultured on LB solid ampicillin medium at 37 °C. Several single colonies were selected and cultured in an oscillator for 14 h at 37 °C and 200 rpm. The plasmid was extracted using the SanPrep Column Plasmid Mini-Preps Kit (Sangon Biotech, Shanghai, China) and was used to amplify the template of *RaSsp1* dsRNA containing the T7 promoter. The PCR product was extracted with phenol–chloroform (1:1) and then was used to synthesize dsRNA using the T7 RNAi Transcription Kit (Vazyme Bio, Nanjing, China). The purity, concentration, and quality of dsRNA were the same as for RNA detection. At the same time, *Egfp* was selected as the control gene, and dsRNA was generated using the *Egfp* vector pQBI-polII (Wako, Osaka, Japan).

Secondly, for the RNA interference experiment, soldiers were anesthetized on ice for 10 min before injection, and 300 nl dsRNA (5000 ng/μL) (*Egfp* or *RaSsp1*) was injected into the soldiers’ abdomen between the second and third thoracic segments using a Model MPPI-3 micro-injector (Applied Scientific Instrumentation, Eugene, OR, USA). The injected soldiers were cultured with new workers at a ratio of 1:3. And then, the samples of the soldiers were collected on each day from the 1st day to the 5th day for analyzing the defensive behaviors of soldiers with the same method mentioned in 2.2 and the gene expression of JH signal key genes with the same method mentioned in 2.3. 20. The soldiers were used as biological replicates, and each replicate included 2 treatments (*RaSsp1* RNAi and control) and 5 time points.

### 2.5. GC-MS

The alarm pheromone is mainly secreted by the frontal gland and released from the fontanelle in the head of termites. To analyze the effect of JH on the alarm pheromone in the soldier heads, the heads of 40 soldiers were crushed using a glass rod, and the tissues were immersed in hexane at a ratio of 10 μL per head and placed in a 4 °C refrigerator for 12 h for extraction. The extracts were centrifuged at 4000 rpm/min for 5 min, and the supernatant was removed for GC-MS. Helium was used as the carrier gas, and the temperature program was set as follows: the initial temperature of 80 °C for 5 min, gradually increased to 220 °C at 10 °C/min for 5 min, then increased to 260 °C at 15 °C/min for 5 min, and finally increased to 320 °C at 15 °C/min for 10 min. The detection substances were completely passed through the HP-5MS non-polar column, and the peak time of different substances was recorded and the content was calculated.

### 2.6. The Effects of Defensive Behavior on Limonene

Due to the fact that more than 60% limonene was decreased in soldiers fed with JHA, observed via GC-MS, limonene was selected to analyze the effect on the defensive behavior in soldiers. A pheromone-stimulated behavioral detection device was designed according to Reinhard, 2002 [29]. A 5 mm diameter hole was drilled into the dish lid (a Petri dish with 3 cm ID) approximately 1.5 cm from the edge of the dish lid. One soldier and three workers were placed into a Petri dish with moist filter paper. A small strip of filter paper (6 mm long and 3 mm wide) can be used to carry 1 μL of the odor source through the hole and introduce it to the Petri dish. The odor source consisted of 1% volume of limonene (TCI, Shanghai, China) with 1 μL of limonene diluted in 99 μL of n-hexane. Therefore, the odor source used as a control was n-hexane. The top half of the strip was fixed to the insect’s needle, and the bottom half of the strip was used to carry pheromones to interfere with termites. For preventing termites to reach the paper strip, the distance between the filter paper and the bottom of the Petri dish was large enough. To reduce the impact of n-hexane, the filter paper strip was left in the air for 5 s after pheromones dropped onto the filter paper, and then the paper strip was fixed on the device together with an ant (the method of fixing ant is the same as detailed in Section 2.2). The defensive behaviors of mandible opening–closing, bites and antennation in soldiers were recorded and analyzed. The experiment was divided into three treatment groups: water, n-hexane, and limonene. Each group consisted of 20 soldiers and 60 workers from five clonal colonies. The frequency of various defensive behaviors exhibited by each soldier within 10 min after exposure to pheromone stimulation was recorded.

### 2.7. Statistical Analysis

All data analysis was performed in DPS 9.01 and GraphPad Prism 9.0.1 statistical software. Student’s *t* test, one-way ANOVA and two-way ANOVA were used to analysis the significance of differences between different treatments. *p* values <0.05 were considered to be statistically significant.

## 3. Results

### 3.1. Defensive Behaviors of Soldier R. aculabialis in Encountering Ants

To investigate the defensive behaviors of *R. aculabialis* soldiers, the abdomen of a *T. caespitum* was glued to an insect pin as an invader at the entrance of the device (Figure 1A). It was observed that workers normally departed quickly and ran across the Petri dish when they perceived a threat. After a worker encountered a soldier, it trembled (jerked its body back and forth), and touched the soldier with its antennae, which may transfer information to the soldier. Following this interaction with the worker, the soldier ran toward the source of intruding ants within the nest. Once a soldier located an ant, it initially bit the invading ant and continued to do so until the ant was dead or injured (Figure 1B1, Appendix A), and then the soldier left after making sure that there was no threat. During the biting process, soldiers often combined to form other defensive postures, including antenna touching with other termites, tremulation, and head-banging (hitting the ground with the head) (Figure 1B2). These behaviors of soldiers may be used to warn ants or signal to other termites.

### 3.2. Effect of JH on the Defensive Behavior of Soldiers

To examine the effect of JH on the defensive behavior of soldiers, we observed and counted the behavior changes in soldiers treated with JHA, including bites, head-banging, mandible opening–closing and antennation, when the soldier encountered an ant. The results indicate that JHA decreased the biting frequency (Figure 2A), increased the frequency of head-banging (Figure 2B), and had no effect on mandible opening–closing and antennation (Appendix A).

After feeding with JHA within 5 days, the biting frequency was higher in the control group compared to either of the experimental groups. There was no significant difference between the 120 μg treated group and the 200 μg treated group. Furthermore, from the 1st to 4th day after feeding with JHA, the frequency of the soldiers’ biting presented a downward tendency, despite there being no significant difference between different feeding days. In contrast to biting behavior, the head-banging frequency of JHA-treated soldiers was significantly higher than that of the control, except for the 120 μg treated group on the 1st day (Figure 2B). The number of head-banging events in the 200 μg treated group was higher than that in the 120 μg-treated group on the first two days. The 120 μg treated group had a significantly higher number of head-banging events from the second day of feeding, which was 1.6-fold higher compared to the control. The 200 μg treated group showed a significant increase in head-banging frequency from the first day of feeding, which was 1.9-fold higher compared to the control group. In addition, from the 1st to 5th day after feeding with JHA, the frequency of soldiers’ head-banging showed a downward tendency, despite there being no significant differences between different days in the three groups.

### 3.3. Effect of JHA on the JH Pathway in Soldiers

We used real-time PCR to analyze the expression changes in *RaSsp1*, *Met* and *Kr-h1* after feeding soldiers with JHA. The results show that the expression of *RaSsp1* was downregulated in the heads of JHA-fed soldiers, particularly in the 120 μg treated group on the 3rd day after feeding, by 61.4% compared with the controls (Figure 3A). The expressions of *Met* and *Kr-h1* were upregulated (Figure 3B,C). *Met* expression increased by 1.9-fold on the 2nd day in the 200 μg treated group compared with the control, and a 3.2-fold increase in *Kr-h1* expression was observed on the 2nd day in the 200 μg treated group compared with the control (Figure 3B,C). In addition, the expression of the *RaSsp1*, *Met* and *Kr-h1* genes in the treatment group tended to be consistent with the control on the 4th day.

### 3.4. Effects of RaSsp1 RNAi on Defensive Behavior in Soldiers

To further evaluate the role of *RaSsp1* in soldiers’ defensive behaviors, the effect of *RaSsp1* silencing on soldiers’ defensive behaviors and the expression of *Met* and *Kr-h1* was analyzed by injecting *RaSsp1* dsRNA into soldiers. The results show that dsRNA injection resulted in the downregulated expression of *Met* and *Kr-h1*, fewer bites and more head-banging events (Figure 4). The expression of the *RaSsp1* gene was significantly downregulated after RNAi treatment, and the interference efficiency was 76% up to the 5th day (Figure 4A). Interestingly, the expression of *Met* and *Kr-h1* was significantly up-regulated on days 1–3 and down-regulated on days 4–5 after RNAi, especially *Met*, which increased 13.6-fold on the 1st day after injection (Figure 4B). The expression of *Kr-h1* increased 2.4-fold on the 3rd day after injection (Figure 4C). After injecting *RaSsp1* dsRNA, the number of bites significantly reduced (Figure 4D), and the number of head-banging events significantly increased (Figure 4E). Of these, the *RaSsp1* RNAi group displayed a 29.5% reduction in the number of bites compared with the controls on the 1st day; the number of bites in the control group ranged from 36 to 69, while that of the experimental group ranged from 21 to 46, with a smaller reduction from the 3rd to the 5th day and 19.0% for the 5th day (Figure 4D). The *RaSsp1* RNAi group also displayed the largest increase in the number of head-banging events on the 1st day after intervention, which was 1.8-fold higher than that of the control group, and the increase was smaller on the 2nd and 3rd days and decreased on the 4th and 5th days compared with the control group (Figure 4E).

### 3.5. Effect of JHA on the Alarm Pheromones of Soldiers

In order to analyze the mechanism of JHA affecting defensive behaviors, GC-MS was used to determine the monoterpenes in alarm pheromones secreted by *R. aculabialis* soldiers. The results show that five pheromones were identified, including α-pinene, β-pinene, limonene (+, −) and two unknown pheromones (peak time 7.3 min and 7.5 min, respectively) (Figure 5A). The concentrations of these five pheromones in soldiers continuously decreased within 5 days (Figure 5B–F); among them, limonene in the 120 μg treated group showed the greatest decrease on day 1, reaching 77.7% (Figure 5D). The concentrations of four pheromones (except for the pheromone at 7.3 min) in JHA-fed soldiers were significantly lower than those of the control, especially on the 1st day. In the following 4 days, the difference between the treated group and the control group gradually reduced. The 120 μg treated group showed the greatest difference on the 1st day, with α-pinene, β-pinene and limonene decreasing by more than 60% compared to the control. The differences between the experimental and control group gradually diminished toward consistency on the 4th and 5th days (Figure 5B–D). There was no significant difference in the effects of two doses of JHA on the concentrations of five pheromones.

### 3.6. Effect of Limonene on the Defensive Behaviors of Soldiers

To further analyze the mechanism of the effect of JH on termite defense behavior, limonene was selected to explore the change in the defensive behavior of soldiers, including bites, mandible opening–closing and antennation (Figure 6). The results indicate that limonene significantly increased the frequency of mandible opening–closing events and bites in soldiers and had no significant effect on antennation. The limonene group displayed a significantly higher number of mandible opening–closing events and bites, with an average number of 13.1 and 21.5, respectively, which was 1.75-fold and 1.71-fold higher than that of the control (n-hexane). There were no significant differences between water and hexane. These findings suggest that limonene releases the defensive behavior of soldiers, though not for all defensive behaviors.

## 4. Discussion

In this study, we exhibited the response of defensive behavior in *R. aculabialis* soldiers to feeding with JHA and the silencing of *RaSsp1* and their effect on the JH pathway genes. We also conducted behavioral experiments to investigate the effects of feeding with JHA on alarm pheromone production and the influence of alarm pheromones on defensive behavior. The results show that JH and *RaSsp1* are involved in the defensive behavior of soldiers and that alarm pheromones serve as a medium for information transmission and participate in this process together. It was interesting that the results of RaSsp1 was opposite to the previous hypothesis. The decrease in *RaSsp1* expression and increase in JH levels caused the same effects on the defensive behavior of soldiers and the expression of *Met* and *Kr-h1*, with JH downregulating the expression of *RaSsp1* and the concentration of alarm pheromones. The changes in soldier termites’ defensive behavior stimulated by alarm pheromones are consistent with those caused by JH and *RaSsp1*. It is speculated that JH might be involved in the defensive behavior of soldiers through the downregulation of *RaSsp1* expression, the upregulation of the expression of *Met* and *Kr-h1*, and the reduced secretion of alarm pheromones (Figure 7).

### 4.1. The Effect of JH on the Defensive Behavior of Soldiers

Social insects have evolved a wide variety of defensive strategies, ranging from nest structures to morphological, physiological, and behavioral adaptations of individuals. Termite soldiers play an essential role in colony defense and cannot reproduce, work on tasks other than defense, or feed themselves. *R. speratus* soldiers effectively blocked access to the nest for termite-hunter ants using a combination of head-plug defense behavior and mandibular threats [2]. The alarm signal transmitted by the alerting termites to their companions is mostly carried out through antenna touching and vibration [30], allowing workers to escape threats by moving away rapidly. In contrast, soldiers are recruited to dangerous locations in order to display defensive postures, often in combination with the secretions of alarm pheromones produced by the labial glands during mandible opening–closing [31,32]. Our results show that *R. aculabialis* workers attempted to escape from intruders by moving quickly and transmitting messages via tremulation/antennation, while soldiers often searched for the source of the disturbance and opened their mandibles to bite consistently until the intruder ant was dead or injured (Figure 1, Appendix A). At the same time, soldiers waved their antennae, engaged in tremulation, banged the ground with their heads, and opened and closed their mandibles (Figure 1B). These behaviors may have been performed to warn off the ants or to send messages to other termites after encountering a threat. The opening and closing of the mandibles of soldiers helps them to be prepared to bite once the opponent reaches their antennae, stimulating the release of defensive chemicals from the cephalic glands, which are usually delivered to the opponent together with the bite [3,14].

JHs, as a hormone related to insect growth and development, is a terpenoid substance that plays a role in regulating insect caste differentiation, diapause, and mating reproduction [33]. Previous studies suggest that JH is involved in the regulation of aggressiveness in some insects, such as cockroaches *N. cinerea* and bees *A. mellifera* [23,25]. Young male aggression in *N. cinerea* was increased with JH injection [23]. JH plays a regulatory role in the levels of aggressiveness in *A. mellifera*, and JH titers of guards are higher than all other middle-aged bees [25]. Topical applications of JH and JHA in termites have been carried out in many studies, such as a high JH titer leading to soldier differentiation [17,20,21,22]. Alarm behaviors are ubiquitous in termites, confirming that caste-dependent responses to disturbances (workers primarily hide away while soldiers confront the threat) are a plesiomorphic characteristic of termites. However, whether JH affects termite defensive behaviors has yet to be reported. It was speculated that JH is involved in the defensive strategy because of its important roles in the morphogenesis of soldier-specific weapons (especially the mandibles) [17,22] and the levels of aggressiveness within *A. mellifera* and *N. cinerea* [23,25]. In this study, this speculation was confirmed by feeding with JHA, restraining bites and encouraging the head-banging of soldiers, which further verified the involvement of JHs in regulating defensive behavior in insects. Therefore, it was speculated that JH inhibits mechanical aggression (bites) and promotes the release of pheromones (head-banging and mandibles opening–closing) [3,31]. Of course, for a more comprehensive explanation of these behaviors and mechanisms, further experiments need to be carried out to confirm whether JH antagonists have the opposite effect on paradigms such as biting and head-banging and whether JHA changes the behavior of an untreated soldier termite in relation to its nest mates.

### 4.2. The Mechanism of JH Regulating the Defensive Behaviors of Soldiers

The nuclear JH receptor Met activated soldier differentiation in *Zootermopsis nevadensis*, and the rapid increase in *ZnMet* expression and the subsequent activation of JH signaling just after the pre-soldier molt were necessary for the formation of soldier-specific weapons [21]. In this study, feeding with JHA causes the upregulated expression of *Met* and *Kr-h1* (Figure 3). The impact of 200 micrograms on the Met gene is greater than that of 120 micrograms, but the behavioral differences are not significant. It is possible that different gene expressions have the same effect on downstream genes.

Another crucial defensive strategy of termites is chemical defense, which enables the recruitment of soldiers and workers to escape the location of threats. The frontal gland, a soldier-specific organ, is a saccular gland that opens to the exterior through the fontanelle in the majority of termites [14]. To sum up, it is speculated that the mechanism of JH regulating the defensive behavior of soldiers involves accelerating the transcription of *Met* and *Kr-h1* to reduce the secretion of alarm pheromones, which then mediates defensive behaviors. Furthermore, it is essential to further research on whether the behavior returns to normal if the experiments are conducted for a few more days, due to the mRNA levels of the affected genes returning to normal levels after 4–5 days of the treatment with a JH analog.

### 4.3. The Cross-Talk between JH and RaSsp1

Based on the similarity of *RaSsp1* to the JHBP domain-containing protein, and its involvement in the formation of soldier-specific morphological characters (i.e., the mandibles and the head capsule), it was speculated that *RaSsp1* affected soldier differentiation from workers by binding and transporting JH [17]. As a gene highly expressed in the heads of soldiers, *RaSsp1* was speculated to be related to their defensive behavior. In this study, *RaSsp1* silencing also affected the defensive behavior of soldiers, leading to a reduced number of bites (Figure 4D) and an increased number of head-banging events (Figure 4E). Interestingly, decreased *RaSsp1* and increased JH caused the same effect on the defensive behaviors of soldiers and the expression of *Met* and *Kr-h1*. It was indicated that JH might be involved in the defensive behavior of soldiers through the downregulation of *RaSsp1* expression. This result contradicts our hypothesis in introduction, so it was speculated that *RaSsp1* affected the defensive behavior of soldiers by other molecular mechanisms than the JH-binding protein. Further research should focus on elucidating the molecular mechanism of *RaSsp1* affecting the defensive behaviors of soldiers.

## Figures and Tables

**Figure 1 insects-15-00130-f001:**
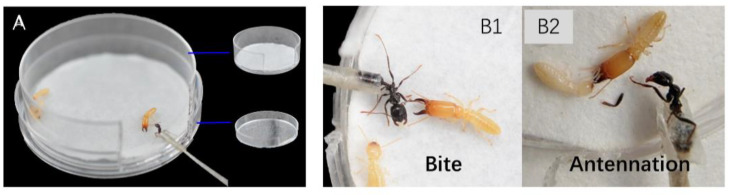
The defensive behavior of termites encountering enemy invasion in *R. aculabialis*. (**A**) Experimental device for quantifying defensive behavior. (**B**) The defensive behavior of soldiers includes bite (**B1**) and antennation (**B2**). Bite refers to the soldiers’ ability to bite predators, while antennation involves the use of their antennae to sense and respond to external stimuli..

**Figure 2 insects-15-00130-f002:**
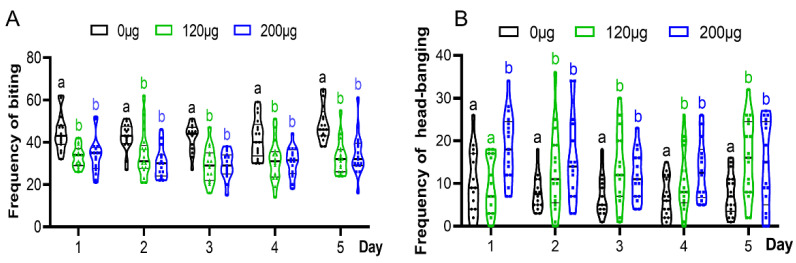
Biting and head-banging frequency of soldiers responding to enemy invasion after feeding with JHA at different times. (**A**) Biting behaviors; (**B**) head-banging (testing sample size was 20 soldiers). Different letters above the bars denote significant differences (one-way ANOVA followed by Scheffe’s F test, *p* < 0.05).

**Figure 3 insects-15-00130-f003:**
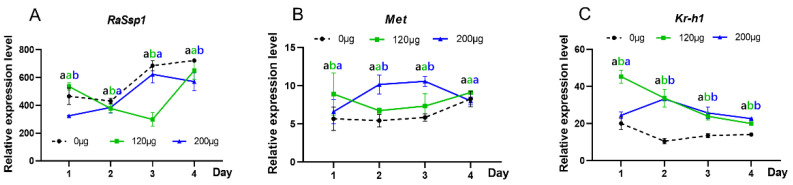
The expression levels of JH signal key genes in soldiers after feeding with JHA. (**A**) *RaSsp1*; (**B**) *Met*; (**C**) *Kr-h1*. Different letters above the bars denote significant differences (one-way ANOVA followed by Scheffe’s F test, *p* < 0.05).

**Figure 4 insects-15-00130-f004:**
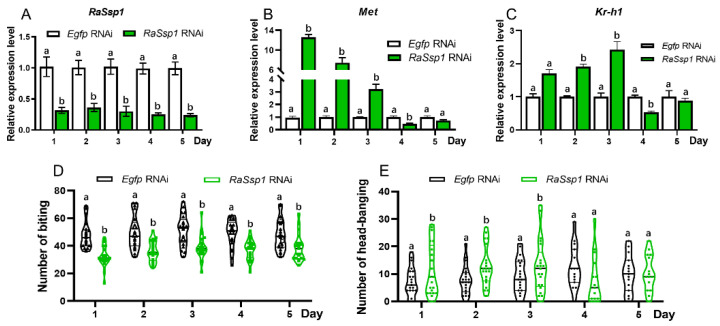
The expression levels of JH signal genes in soldiers and the change in defensive behavior after *RaSsp1* RNAi. (**A**) *RaSsp1*; (**B**) *Met*; (**C**) *Kr-h1*; (**D**) bites; (**E**) head-banging. Different letters above the bars denote significant differences (one-way ANOVA followed by Scheffe’s F test. Lower case letter shows *p* < 0.05). Black and green columns indicate *Rassp1* and *Egfp* dsRNA-injected individuals, respectively.

**Figure 5 insects-15-00130-f005:**
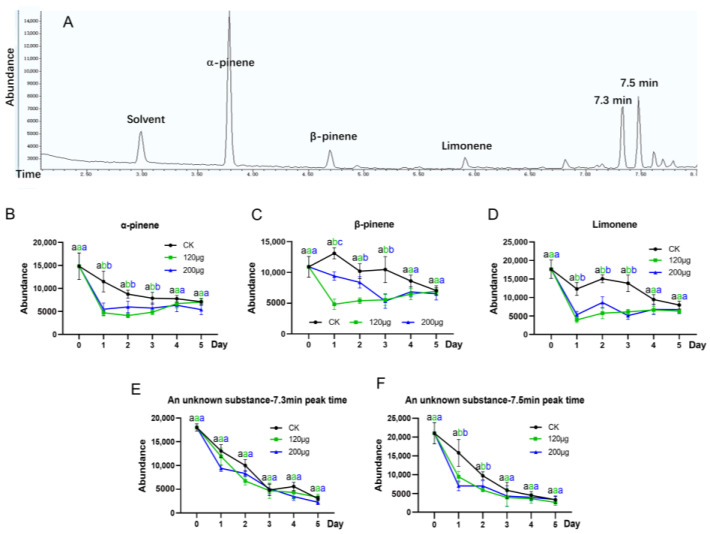
The abundance of alarm pheromones in soldiers after feeding with JHA. (**A**) Total ion chromatogram of defense secretions extracted from *R. aculabialis* soldiers’ head; (**B**) abundance of α-pinene; (**C**) abundance of β-pinene; (**D**) abundance of limonene (+, −); (**E**) abundance of unknown substance—7.3 min in peak time; (**F**) abundance of unknown substance—7.5 min in peak time. Error bars represent mean values ± SEM. Moreover, 40 soldiers were taken for each treatment group for hexane extraction. Different letters above the bars denote significant differences (one-way ANOVA followed by Scheffe’s F test, *p* < 0.05).

**Figure 6 insects-15-00130-f006:**
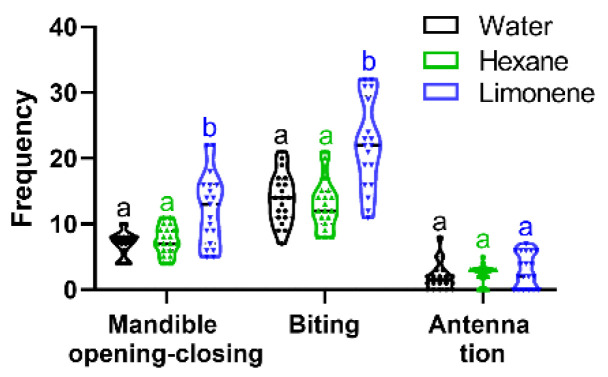
Effect of limonene on the frequency of defensive behavior in soldiers. Different letters above the bars denote significant differences (one-way ANOVA followed by Scheffe’s F test, *p* < 0.05).

**Figure 7 insects-15-00130-f007:**
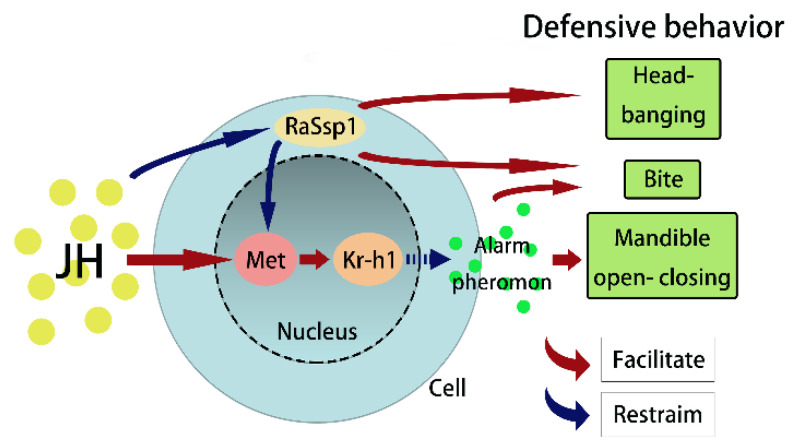
The mode of juvenile hormone involved in the defensive behaviors. Arrows denote the flow of signaling molecules and their interactions within the JH pathway, elucidating the regulatory roles of JH and RaSsp1 in defensive behavior.

## Data Availability

All data generated and analyzed in this study are included in the published article and its Appendix A.

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
