# Peer review of "Juvenile Hormone Involved in the Defensive Behaviors of Soldiers in Termite Reticulitermes aculabialis"

_insects, 2024, doi:10.3390/insects15020130_

Round 1

Reviewer 1 Report

Comments and Suggestions for Authors

This study by Li et al, studies the role of JH in defense behavior of soldiers in the Reticulitermes aculabialis termites. They reported that feeding termites with JH analogs decreases biting and increases head-banging in soldiers while also downregulating RaSsp1Met and Kr-h1. This was consistent with RaSsp-1 knock-down causing the same behavioral outcomes, besides causing a decrease in the secretion of several pheromone components out of which limonene affects defensive behavior. The experiments are thoroughly conducted with proper controls and the flow of experiments has a well-defined logical pattern. The results strongly support the conclusions made by the authors. The findings of the study will be of interest to the readers of this journal. However, some minor issues need to be addressed:

1. The authors could extract more information from the behavioral data that they already have. Some more information about whether the latency of induction of behavior or the sequence of behavioral displays are affected will be great to have.

2. mRNA levels of the affected genes and even pheromone secretion returns to normal levels after 4-5 days of treatment with JH analog or RaSsp1. However, the behavioral changes still persist. It is essential to test if the behavior returns to normal if the experiments are conducted for a few more days. Also, the authors need to address this in the discussion.

3. It will be worthwhile to test if JH-antagonists may have an opposite effect on paradigms such as biting and head-banging.

4. The authors need to have more details of the functional utility of the head-banging behavior.

5. It appears JH  not only changes defensive behavior in soldiers but also, modulates other nest-mates to behave in a ceratin way. Hence it will be essential to test whether presence of a JHA fed or RaSsp knocked-down termite can change the behavior of an untreated soldier termite 

Comments on the Quality of English Language

English is mostly fine with some random sentences being erroneous. Although they don't interfere with comprehension, at times they are irritating. The authors need to correct those.

Reviewer 2 Report

Comments and Suggestions for Authors

The main results reported in this article are the effects of pyriproxyfen treatment on the defensive behaviors of termite soldiers and the effects of knockdown of expression of the gene that encodes RaSsp1 on the same defensive behaviors. The direct rationale for the pyriproxyfen treatment experiments is a small prior literature on JH and defensive/aggressive behaviors in other insect species. There is also an indirect rationale along the lines of "if JH is important in differentiation of the soldier caste in pre-adult stages, perhaps it also regulates adult physiology and behavior."  This is not a new idea, and in fact I was surprised that the did not cite the important 1997 review by Robinson and Vargo that built a case for this concept. (Robinson GE, Vargo EL. Juvenile hormone in adult eusocial Hymenoptera: gonadotropin and behavioral pacemaker. Arch Insect Biochem Physiol. 1997;35(4):559-83). The rationale for the RaSSp1 gene dsRNA experiments is bioininformatics support for the function of RaSsp1 as a JH binding protein, although as far as I am aware there is no direct evidence for this function. The effects of these manipulations on defensive/aggressive behavior were assessed in a very interesting laboratory (Petri dish) assay using a tethered ant. Video recordings of the assay allowed the researchers to count the number of bites, head-bangings, openings of the mandibles, and antennations. The effects of pyripoxyen on expression of RASsp1, the JH receptor Met, and the JH-induced early gene Kr-hi were also reported to support interpretation of the results. Data from a set of marginally-related experiments on alarm pheromone were also presented. The paper will interest a relatively small audience of readers , as this is not currently an active area of research in insect behavior. Potential readers, however, will be discouraged by the confusing nature of the presentation. My specific comments will focus on suggestions to make this paper less confusing.

1. One way to make the paper significantly less confusing would be to present the model introduced in the Discussion (Figure 7) in the Introduction and to use it to make specific predictions that the authors' experiments will test. Based on my reading of the paper and Figure 7, I think that the authors predict that treatment of termites with a JH analogue will modify the behavior of termite soldiers in a laboratory assay of aggressive/defensive behavior. I think that their second prediction is that reduction of RaSsp1 expression will have similar effects, because a component of their model is that RaSsp1 (presumably because it binds JH) reduces activation of JH pathways, so that reducing RaSsp1 expression will release the inhibition imposed by the binding protein. Organizing the Introduction around the hypotheses to be tested would significantly improve this paper. I think it would also help the authors conclude that the material related to alarm pheromones should be removed from this paper.

2. A second way to make the paper significantly less confusing would be to do a better job explaining the experimental design. In particular, the authors should justify the experimental treatment choices. For example, why were the doses of 120 and 200 micrograms selected, and how much of the filter paper did the termites actually consume? The paper cited [14] is not useful because it deals with development, not adult behavior. Why were the experiments conducted for five days? Were soldiers tested repeatedly. or were fresh soldiers tested on each day? Did treatment with pyripoxyfen occur just one time at the start of the experiment or was it continued across the five days? Why was pyripoxyfen selected, as opposed to hydroprene or methoprene? What was the impact of treatment with pyripoxyfen on survival? Why was a "no treatment" group not included? The absence of a baseline no treatment group means that it is possible that the observed results pyripoxyfen treatment on behavior reflect exposure to acetone and not the effects of the JHA.  Another detail that should be included is the time of day that testing occurred.

3. The authors could also improve this paper by paying more attention to the behavior they are studying. Why did they select such a small number of behaviors? Why did they rely entirely on frequency data and not use other measures such as latency, time to killing of the ant (duration of the attack)? And why - and to me this is a missed opportunity - why do they not discuss in more detail their finding that there is no such thing as global defensive behavior, given that their treatments so clearly impacted the various behaviors in different ways. I think this could have led to a very interesting discussion. I was surprised that biting and mandible opening-closing were differentially impacted. If the goal of studying behavior is to understand the neural pathways that produce behavior and how they are modified by hormones, a fuller description of the behavior would be a good starting place. More literature citations on soldier behavior would also be helpful.

4. There is not a particularly good temporal correlation between the effects of the treatments on gene expression and the changes in behavior. The authors also did not discuss how they interpret the different responses (in terms of gene expression, Figure 3) to the 120 and 200 microgram doses of pyripoxyfen.

5. Other suggestions. Change the title to "Pyripoxyfen defensive behaviors of soldiers of the termite Reticulitermes aculabialis" because this is what is studied. Make Figures 3 and 4 easier to read by restricting the Y-axis to cover the range of the reported values. Do not present the same information from the literature in the Introduction and Discussion sections.

In summary, I think this paper is interesting and I really like the Petri dish assay but in my opinion, to be impactful, the paper needs a presentation focused on experimental design and behavior. 

Comments on the Quality of English Language

There are many minor errors that an editor could detect. A misunderstanding, however, could arise from lines 83-84. As currently written the statement implies that JH may regulate defensive behavior in male honey bees. The authors should mention honey bees first in this sentence and also specify that the reported effects were detected in female workers.
